# Safety and Regenerative Properties of Immortalized Human Mesenchymal Stromal Cell Secretome

**DOI:** 10.3390/ijms26199322

**Published:** 2025-09-24

**Authors:** Maxim Karagyaur, Alexandra Primak, Nataliya Basalova, Anna Monakova, Anastasia Tolstoluzhinskaya, Maria Kulebyakina, Elizaveta Chechekhina, Mariya Skryabina, Olga Grigorieva, Vadim Chechekhin, Tatiana Yakovleva, Victoria Turilova, Elena Shagimardanova, Guzel Gazizova, Maksim Vigovskiy, Konstantin Kulebyakin, Veronika Sysoeva, Uliana Dyachkova, Stalik Dzhauari, Kirill Bozov, Vladimir Popov, Zhanna Akopyan, Anastasia Efimenko, Natalia Kalinina, Vsevolod Tkachuk

**Affiliations:** 1Medical Research and Education Institute, Lomonosov Moscow State University, 27/1, Lomonosovsky Ave., 119192 Moscow, Russia; 2Institute of Cytology, Russian Academy of Sciences, Tikhoretsky Ave., 4, 194064 St. Petersburg, Russia; 3Skolkovo Institute of Science and Technology (Skoltech), Genomics and Bioimaging Core Facility, Skolkovo Innovation Center, Bolshoy Boulevard, 30, Bld. 1, 121205 Moscow, Russia; 4Life Improvement by Future Technologies Institute, Nobel Str., 5, 121205 Moscow, Russia; 5Institute of Fundamental Medicine and Biology, Kazan Federal University, 420008 Kazan, Russia

**Keywords:** cell secretome, immortalized cell cultures, mesenchymal stromal cells, regeneration, Leydig cell culture potency assay, fibroblast-to-myofibroblast differentiation

## Abstract

The secretome of mesenchymal stromal cells (MSCs) can efficiently stimulate regeneration and therefore is a tempting remedy for “cell-free cellular therapy”. However, the usage of primary MSC cultures as secretome producers for translation studies has obvious obstacles, including the rapid aging of MSC cultures, the need for a large number of verified donors, and donor-to-donor variability of secretome content. MSCs immortalization makes it possible to overcome those limitations and to obtain secretome-producing cultures with a prolonged lifetime. However, the efficacy and safety of such secretomes are critical issues that limit their usage as therapeutic agents. In this study, we tested in large detail how the immortalization of MSC cultures affects the content, biological activity and safety of their secretome. MSCs immortalization via the overexpression of human *TERT* gene does not significantly alter the qualitative and quantitative composition of their secretome or its activity according to the results of proteomic analysis, ELISA, qPCR and functional tests in vitro. Moreover, we have demonstrated that the secretome of immortalized MSCs does not contain detectable amounts of telomerase and does not possess any transforming activity. Altogether, our data suggest that immortalized MSC cultures may become a reliable source for obtaining standardized active secretome in large-scale quantities for clinical use.

## 1. Introduction

Regeneration of damaged organs and tissues is a complex and multi-component process that involves a large number of molecular and cellular participants [1,2]. This largely determines the limited capacity for spontaneous regeneration within the human body [3]. However, the regeneration of some organs can be stimulated by supplying exogenous molecules able to slow down the damage progression and to launch the proliferation and differentiation of stem cells. Potentially, the role of such exogenous molecules may be filled by recombinant growth factors, cytokines, matrix proteins, or certain pro-regenerative microRNAs [4,5,6]. Meanwhile, a number of shortcomings limits the widespread usage of recombinant molecules in clinical regenerative medicine. These shortcomings are high production costs, the possibility of side or toxic effects during prolonged usage, and, most importantly, the ability of such molecules to affect only certain restricted cellular and molecular targets within the processes of tissue damage and regeneration [7,8]. Altogether, this reduces their therapeutic effectiveness.

A more effective approach to stimulating tissue regeneration is a combined therapeutic effect using factors that functionally complement one another [9,10,11,12]. For this purpose, both combinations of recombinant proteins or molecular compositions based on the secretion products of progenitor and stem cells can be used [13,14]. Particular attention in this context is paid to the possibility of using the secretion products of multipotent mesenchymal stromal cells (MSCs) that are considered to be the natural coordinators and stimulators of tissue renewal and regeneration [15,16].

Numerous studies have shown that MSCs produce a wide range of biologically active molecules, including neurotrophic and proangiogenic growth factors, proteases and matrix proteins, coding and regulatory RNAs [17,18,19,20]—altogether comprising the MSC secretome. Due to the presence of functionally complement molecules, the secretome also exerts a combined therapeutic effect on the processes of tissue damage and regeneration, including the suppression of apoptosis, fibrosis and excessive inflammation, the stimulation of recruitment and differentiation of tissue-specific stem cells, the restoration of blood supply and the innervation of the regenerating tissue [20,21]. In most cases, the MSC secretome can completely mimic the pro-regenerative activity of MSCs themselves [22,23,24,25,26]. At the very same time, the secretome is free from the shortcomings of traditional cell therapy. Particularly, its use is associated with a lower risk of immunological, vascular, and tumorigenic complications that are common for cell therapy [27,28,29,30,31]. Unlike cell-based medicines, cell-free secretomes do not require special storage and transportation conditions [32] and can be used “off-the-shelf” for the treatment of urgent pathological conditions (e.g., acute brain injuries) [33]. All this allows us to consider the secretomes of progenitor and stem cells (in particular, MSCs) as a convenient platform for the development of promising drugs for regenerative medicine.

Like with any other therapeutic approach, the usage of secretomes of primary cell cultures (and MSCs as well) has a number of limitations that need to be overcome for the widespread usage of this technology to become possible. They is the need for a large number of donors, source- and donor-dependent variability of MSC and secretome properties [34,35,36,37], rapid aging of an MSC culture (up to 7–9 passages) and the necessity of biological and microbiological characterization of each new MSC culture [38,39]. All this significantly increases the cost of obtaining such a cell-free composition and complicates its standardization [40,41,42].

One approach to overcome these limitations is the usage of standardized immortalized MSC (iMSC) cultures capable of maintaining their proliferative activity for up to 40 and even more passages [42,43], which can be largely expanded and used for secretome production. According to the literature, immortalization via the overexpression of the telomerase reverse transcriptase catalytic subunit gene (*TERT*) allows cell cultures to retain functional p53 and pRB checkpoints in mitosis, normal cell cycle control, intrinsic karyotypes and phenotypes, sensitivity to contact inhibition, dependence on growth factors and surface adhesion [44,45], which makes them radically different from transformed/tumor cell cultures. However, the influence of the immortalization procedure on the stability of MSC secretome composition, its pro-regenerative properties, as well as its potential tumorigenicity has practically not been studied, which stalls its potential translation to clinical practice.

We hypothesized that the secretome of TERT-immortalized MSCs could represent a convenient and easily scalable platform for the development of drugs for regenerative medicine. The higher proliferative activity of an immortalized cell culture makes it possible to obtain more producer cells, and therefore larger volumes of secretome with low or no batch-to-batch variability [42,46,47]. Potentially, this could lead to increased efficiency, accessibility and reduced cost of regenerative technologies, and their faster translation into clinical practice. Therefore, in this study we have tested in large detail how the immortalization of an MSC culture via *TERT* overexpression affects the qualitative and quantitative content, its biological activity in validated in vitro models and the safety of its secretome.

## 2. Results

### 2.1. Immortalization Stabilizes the Qualitative and Quantitative Composition of the MSC Secretome

To evaluate how the immortalization procedure changes the qualitative composition of the secretome, we performed a proteomic analysis of the secretomes of primary and TERT-immortalized MSCs (pMSC and iMSC secretomes, respectively). A total of 1338 proteins were identified: 1207 proteins in the pMSC secretome and 1214 proteins in the iMSC secretome. The secretomes of pMSCs and iMSCs were 94.5% identical. They included neurotrophic factors, neuropoietic cytokines and peptides (GMFB, MANF, MEGF8, NDNF, NENF, Neuroserpin, OLFM1, IL-6, LIF); proangiogenic factors (VEGF-A, uPA, tPA, PlGF, PDGF, angiopoietin, angiopoietin-like proteins-2/-4, follistatin, inhibin A, midkine, MMP-2/-9, TIMP-1/-2, leptin, VASN); anti-inflammatory cytokines (TGFb, GDF15, MYDGF); growth factors (bFGF, EGF, IGF-1/2, KGF, SDF-1, neuregulins, TCN2, NEGR1, HDGF, CTGF, PEDF, and insulin-like growth factor binding proteins); matrix proteins (fibronectin, collagen-1a1/-4a1, laminin-a2/b2, periostin, neuroligin-2, nidogen-1/-2); heat shock proteins (HSP70, HSP74, HSP90B1, etc.), as well as molecules involved in the process of sorting and packaging of microRNAs (including neuroprotective ones) into extracellular vesicles (HNRNPU) [48,49].

A number of proteins identified in the secretomes of both primary and iMSC represent intracellular proteins of MSCs (cytoplasmic, nuclear, ribosomal, mitochondrial and membrane). These proteins become a part of the cell secretome due to MSC death during conditioning and to protein inclusion in extracellular vesicles, which are an integral part of the cellular secretome. Their potential roles in stimulating processes of tissue regeneration have not been unequivocally established to date.

Minor differences were found between secretomes of primary and iMSC by the content of molecules with proven protective and pro-regenerative activity. Thus, LIF and VEGF-C were found only in the pMSC secretome, while RFTN1 (extracellular vesicle protein) and olfactomedin like 3 (OLFML3), a secreted matrix protein with proangiogenic properties, were unique for the secretome of iMSCs. In the iMSC secretome, no signs of telomerase were detected.

The limited funding for this study did not allow for a large number of replicates (n = 1), which prevents us from performing statistical analyses and quantitatively assessing changes in the expression of a wide range of proteins after immortalization. However, the data obtained allows us to conclude that the qualitative composition of the MSC secretome remains preserved after immortalization. The raw data of the proteomic analysis has been deposited in the PRIDE repository [https://www.ebi.ac.uk/pride/archive/projects/PXD067843, accessed on 21 September 2025].

To evaluate how immortalization changes the expression of certain growth factors and cytokines, involved in tissue regeneration, we have performed qPCR and ELISA tests of the pMSC and iMSC secretomes. Secretome ELISAs for key neurotrophic and proangiogenic molecules (BDNF, VEGF, uPA and HGF) revealed that immortalization caused no significant quantitative changes in these growth factors within the iMSC secretome (Figure 1A, n ≥ 3). The concentration of these growth factors within the iMSC secretome also did not significantly change when passaged from passages 9 to 42 (Figure 1B, n ≥ 3).

A qPCR analysis of the mRNA of a wider range of neurotrophic and proangiogenic molecules (*BDNF*, *IL-6*, *uPA*, *VEGF*, *HGF*) and key pro-inflammatory (*IL-1A*, *TNFa*) and anti-inflammatory (*IL-4*, *IL-10*, *IL-13*, *TGFb1*, *IDO*) molecules also revealed no statistically significant changes in their expression in MSCs after immortalization (Figure 1C, n = 3). No significant changes in the mRNA expression level of the tested growth factors and cytokines were also established in iMSCs when passaged (Figure 1D, n = 3). The obtained results of mRNA expression levels for most factors correlated well with the ELISA results described above; however, a discrepancy for *HGF* at 12–16 and 25–33 passages was detected.

According to qPCR results, the *HGF* mRNA expression at passages 12–16 and 25–33 increased by 2.7 and 3.2 (compared to the passage 9), respectively, whereas the level of HGF protein according to ELISA data from passages 5–9 to 41–44 remained almost unchanged. Statistical analysis showed that there were no significant differences in the *HGF* mRNA expression level between the iMSC-#2 groups from passages 5 to 44 (one way ANOVA, pairwise multiple comparison, Holm–Sidak method), and the observed discrepancy was explained by the small sample power (n = 3).

### 2.2. SASP Components Appear in the iMSC Secretome Later than in the Secretome of pMSCs

To evaluate how the immortalization procedure changes the onset of expression of the secretome components associated with cell culture senescence (the so-called SASP components), MCP-1, PAI-1 and IL-6, we conducted ELISA tests of the pMSC and iMSC secretomes. Overexpression of these molecules usually coincides with a pro-inflammatory and impaired pro-regenerative potential of the secretome.

We found that the iMSC secretome contains significantly less SASP components than the pMSC secretome at comparable passages, at least up to 18 passages (Figure 2, *p* < 0.05, n ≥ 3). With increasing passage, we observed an increase in MCP-1, PAI-1, and IL-6 content in the secretomes of both pMSC and iMSCs. In particular, in the secretome of iMSCs at the 19th passage, MCP-1 increased up to 2.78 ± 0.52 pg/mL/1000 cells, which is higher than in the secretome of pMSCs at passages 3 to 14. Meanwhile, the significant increase in IL-6 and PAI-1 content was observed only after passage 19. An increase in all three major SASP components in the secretome of iMSCs was observed at passages 25–33, with MCP-1—3.56 ± 0.77 pg/mL/1000 cells, PAI-1—5.69 ± 1.37 pg/mL/1000 cells, and IL-6—0.99 ± 0.14 pg/mL/1000 cells.

The dynamics of the SASP protein increase in the MSC secretome is specific up to the late passages and is not due to culture aging from prolonged (7 days) conditioning, as it is observed only at later passages in pMSCs and iMSCs (and not for all passages of iMSC cultures). For primary MSCs, the finding that the dynamics of SASP proteins increase in the secretome corresponds to previously published data [50,51].

### 2.3. The iMSC Secretome Reveals the Same Pro-Regenerative Activity as the Secretome of pMSCs

To assess the pro-regenerative potential of the secretome of iMSC cultures, we used three previously described diverse in vitro models, able to determine the therapeutical potential of biological compositions. They are: (1) testosterone production by Leydig cells [52,53]; (2) TGF-b induced fibroblast-to-myofibroblast differentiation [54,55]; (3) mouse sensory ganglion neurite growth [56,57]. The Leydig cell-based and fibroblast-to-myofibroblast differentiation models were previously validated for assessing the pharmacological activity of biological compositions for regenerative medicine, to treat fibrosis and male infertility, respectively [52,53,54,55]. The Leydig cell-based model is used for MSC secretome quality control within clinical trials [No. NCT06869863, https://clinicaltrials.gov/study/NCT06869863, accessed on 21 September 2025]. Although these models are cellular, they reflect the biological activity of drugs (they are potency assays), in particular the secretome of MSCs, as pro-regenerative drugs in in vivo models [52,53,54,55,56,57].

#### 2.3.1. iMSC Secretome Stimulates Secretory Activity of Leydig Cells

The secretome of primary human MSCs was previously shown to restore spermatogenesis in animal models of cryptorchidism and chemotherapeutic damage to spermatogenesis [52]. Testosterone secretion by Leydig cells turned out to be one of the mechanisms of pMSC secretome activity in those models. Based on this, we have developed an in vitro MSC secretome potency assay model in which the pMSC secretome significantly stimulates testosterone secretion by Leydig cells at day four, compared to negative controls [52]. Here, we used this verified in vitro model to assess the potency of the iMSC secretome in maintaining the secretory activity of Leydig cells.

The results obtained allow us to assume that Leydig cells stimulated by the secretome of iMSCs produce equal testosterone compared to Leydig cells stimulated by the pMSC secretome (Figure 3A). Moreover, the iMSC secretome retains its activity even when being obtained from iMSCs at passages 12–18 and 25–33, although we detected a tendency for its activity to decrease with passages (Figure 3B).

#### 2.3.2. iMSC Extracellular Vesicles Prevent Fibrosis in the Model of Fibroblast to Myofibroblast Differentiation In Vitro

We have previously shown that extracellular vesicles of primary MSCs (MSC-EV) may prevent the TGFb-induced fibroblast-to-myofibroblast differentiation [55,56]. Using this model, we tested whether extracellular vesicles (EV) from iMSC retain this ability at various passages. The MSC-EV obtained here have not been fully characterized, as the focus of our research was different. Here, we rely on the previously published results of our colleagues [58,59,60], who thoroughly compared the composition and properties of EV obtained from primary and TERT-immortalized MSC cultures. They demonstrated high similarity in size, composition, and properties of these two sets of MSC-EV. The results of these studies established that TERT-immortalization does not critically affect the composition and properties of EV produced by MSCs. The Nanoparticle Tracking Analysis (NTA) conducted by us revealed that pMSC and iMSC secretomes do not differ significantly in the concentration and size of EV (Appendix A).

Using Western blotting, we confirmed that the adding of EV from primary MSCs does not lead to an increase in the amount of alpha-actin protein during differentiation, compared with the positive control group. Using EV from immortalized cells has a similar effect (*p* < 0.05, n = 3, Student’s *t*-test). The important result is that EVs retained their antifibrotic potential even being obtained from iMSC cultures at later (25–33) passages. We also confirmed that in all cells with the addition of EV, the number of stress fibrils with embedded aSMA was significantly reduced (Figure 4). For a separate visualization of aSMA and total actin expression (phalloidin staining), please, see Appendix A.

#### 2.3.3. iMSC Secretome Stimulates Neurite Growth of Murine Sensory Ganglions

The secretomes of primary and TERT-immortalized MSCs reveal a prominent ability to stimulate the growth of nerve fibers of an explant culture of murine sensory dorsal root ganglions (DRG). Data analysis showed that the surface area occupied by the nerve fibers of murine sensory neurons in the groups, where pMSC and iMSC secretomes were applied, was 1.91 ± 0.45 and 2.2 ± 0.49 folds, respectively, significantly larger than the same parameter in the control group (DMEM/F12) 20 days since the start of the experiment (*p* < 0.05, n ≥ 5, ANOVA Holm–Sidak test)—see Figure 5. The identity of the observed microscopical structures with nerve fibers was confirmed in some samples by immunohistochemical staining with antibodies to the neuron-specific heavy neurofilament protein NF200 (Appendix A). The data obtained correlates with the results of testing the neuroprotective activity of pMSC and iMSC secretomes in an in vivo model of intracerebral hemorrhage in rats [61].

### 2.4. The Secretome of Immortalized MSCs Does Not Contain Detectable Amounts of Telomerase and Reveals No Transforming Activity

To assess the potential transforming activity of the iMSC secretome, we have evaluated: (1) the presence of telomerase in the iMSC secretome as a protein or telomerase-encoding nucleic acid (DNA, RNA); (2) the ability of iMSC secretome to initiate colony growth in a culture of primary isolated dermal fibroblasts in soft agar colony formation assay; (3) the transcriptomes of primary isolated dermal fibroblast cultures (the expression changes in oncogenes and anti-oncogenes, above all) treated with the secretome of pMSCs or iMSCs.

#### 2.4.1. The iMSC Secretome Does Not Contain Detectable Amounts of Telomerase Protein or Telomerase Encoding Nucleic Acids

Potentially, telomerase reverse transcriptase being transferred with the secretome of immortalized cells may exert undesirable transforming (malignization) activity on target cells. If so, this may impede its potential translation into clinical practice. To study this, we examined the iMSC secretome for the content of telomerase protein or telomerase encoding nucleic acids. To increase the sensitivity of the methods used, secretomes of pMSCs and iMSCs were concentrated 100-fold. The content of telomerase protein or telomerase encoding nucleic acids (DNA and RNA) in 100-fold concentrated iMSC secretome was found to be lower than the sensitivity of the used detection methods (Western blotting and qPCR) (Figure 6A). Even after exposure for 300 s with Clarity™ Western ECL Substrate on the polyvinylidene difluoride (PVDF) membrane, stained with antibodies to human TERT, no specific signal was detected (Figure 6A and Appendix A). The functionality of the antibodies was confirmed by their ability to stain the catalytic subunit of telomerase in cell lysates (Appendix A).

#### 2.4.2. iMSC Secretome Does Not Induce Fibroblast Colony Formation in Soft Agar Colony Formation Assay

No ability of primary human dermal fibroblasts to form colonies in the soft agar colony formation assay upon iMSC secretome application was observed. In all experimental groups (pMSC and iMSC) single cells and cell duplets were observed. Cell duplets arose from dividing cells that were in mitosis at the moment of seeding. Multicellular colonies consisting of more than four cells were detected only in 0.01% dimethyl sulfate and three mM sodium nitrite groups (positive control groups). The data obtained demonstrate the validity of this test and the absence of transforming activity of the secretomes of primary and immortalized MSCs.

The samples of the micrographs of single cells and cell colonies as well as the result of analysis of their distribution in control and experimental groups are shown in Figure 6B.

#### 2.4.3. The iMSC Secretome Does Not Change the Expression of Pro- and Anti-Oncogenes in the Culture of Primary Human Dermal Fibroblasts

Transcriptomic profiling of lysates of human dermal fibroblasts treated with secretomes obtained from primary and immortalized MSC cultures revealed that the iMSC secretome did not significantly affect the expression of 99.7% of genes (19,279 out of 19,336), including potential oncogenes and crucial tumor suppressor genes, compared to fibroblasts exposed to the secretome of primary MSCs (Figure 6C, n = 2). Special focus was given to the expression levels of potential oncogenes including *BTK*, *EGFR*, *ERBB2*, *FLK1*, *FLT1*, *FLT4*, *HRAS*, *KRAS*, *MYC*, *MYCL*, *MYCN*, *NRAS*, *PDGFR*, *RAF1*, *SRC*, *WWTR1*, and *YAP1*, as well as tumor suppressor genes such as *APC*, *BRCA1*, *BRCA2*, *CDKN2A*, *DENND2B*, *FAS*, *NF1*, *PTCH*, *RB1*, *ST7*, *ST14*, *TP53*, *VHL*, and *YPEL3*.

In the positive control groups (sodium nitrite and dimethyl sulfate), 0.6% and 0.8% (116 and 154 out of 19,336, respectively) of genes changed their transcriptional activity, respectively; however, no pro- and anti-oncogenes involved in cell malignant transformation were identified among them (Figure 6C). The raw sequencing results obtained were deposited in the GEO repository [https://www.ncbi.nlm.nih.gov/geo/query/acc.cgi?acc=GSE306748, accessed on 21 September 2025].

The observed discrepancy between the transcriptome analysis and the soft agar colony formation assay results may be attributed to the low proportion of fibroblasts undergoing transformation in response to sodium nitrite or dimethyl sulfate exposure, as well as the relatively low abundance (below the sensitivity of the method) of pro- and anti-oncogene transcripts in cell lysates.

## 3. Discussion

This study provides extensive details of the assessment of the biological activity and safety of MSC immortalized by telomerase overexpression. The idea of using immortalized cells and their secretome to stimulate regeneration is not original. Thus, Park et al. (2016–2020) demonstrated the ability of the secretome of immortalized murine heart stem cells to stimulate cardiomyocyte protection and reduce the lesion focus in an in vivo model of myocardial infarction [62,63]. Kraskiewicz et al. showed that immortalized adipose tissue-derived MSCs stimulated the healing of chronic venous ulcers via stabilized production of growth factors [64]. The approach we propose here corresponds to the world trend of looking for the reliable source of the agent for cell-free cellular therapy. Despite the number of such studies gradually increasing, the effect of cell immortalization on the composition, stability and safety of their secretome remain largely unexplored.

According to some studies, immortalization may disturb the qualitative or quantitative composition of a cell secretome. Thus, Park et al. observed a decrease in the VEGF and IL-6 content in the secretome of immortalized heart stem cells, despite the fact that the secretome itself retained cardioprotective properties [62]. To study this aspect of immortalization on the spectrum and quantity of produced growth factors, we compared secretome proteomes, the expression and production levels of some key pro-regenerative factors in the secretome of MSCs before and after immortalization. The data obtained allow us to assert that MSC immortalization has not significantly changed the qualitative and quantitative composition of MSC secretome. Expression of human *TERT* delays the appearance of SASP components in the secretome of MSCs. Moreover, immortalized MSC cultures retain the initial level of BDNF, VEGF, HGF and uPA production, as well as the mRNA expression of these and other factors during passaging at least up to 25–33 passages (depending on the iMSC culture). The results obtained here diverge somewhat from the data of Park et al. [62]. Apparently, the conclusion about the effect of immortalization on the composition of the cell culture secretome may be determined by the properties of the cell culture, the immortalization approach used, cell culture passages and the spectrum of factors studied.

The immortalization of an MSC culture not only stabilizes the composition of its secretome but preserves its pro-regenerative ability as well. Thus, we have shown that the iMSC secretome (even obtained from iMSC at 30th passage) prevents fibroblast-to-myofibroblast differentiation (an in vitro model of fibrosis), stimulates secretory activity of Leydig cells in vitro and stimulates the neuritogenesis in the explant model of murine DRG, which completely corresponds to the previously described activity of pMSC secretome at early (5–9th) passages [52,55,65]. Therapeutic activity of iMSC secretome was also previously demonstrated in the model of intracerebral hemorrhage in rats [61].

The preservation of the biological activity of the iMSC secretome at later passages (more than 30th) requires additional studies. There are reasons to suppose that, at later passages, the biological activity of MSC secretome may decrease. Our previously published study characterizing the culture properties of the iMSC cultures obtained demonstrates that *TERT* overexpression increases the iMSC proliferative potential and postpones their aging; however, it does not make them truly immortal [66]. That is manifested in a significant slowdown of their proliferation and cell culture senescence at passages later than the 30th. However, even increasing the period of active use of a cell culture from 9 to 30 passages theoretically makes it possible to obtain 3^(30-9)^ or 3^21^ (three sextillion) more cells and secretome, which makes the standardization of a potential pro-regenerative drug real, minimizes its production costs and brings its clinical application closer. It is worth noting that freezing the iMSCs does not change the composition and biological activity of the secretome produced by them.

One of the current limitations in the large-scale use of immortalized cell secretomes in the clinic is the assumption that such a secretome may contain telomerase and exert transformative pro-malignant activity upon the introduction into the body. This assumption looks logical in light of the fact that transformed (tumorigenic) cell cultures are able to transform their microenvironment and make it pro-malignant/pre-metastatic, due to the secretion of growth factors and microRNAs [18,67,68,69,70]. In order to assess such an undesirable activity of the iMSC secretome, we evaluated its telomerase content and potential ability to alter the expression of pro-oncogenes and anti-oncogenes in target cells (primary fibroblasts), as well as its ability to stimulate fibroblast malignization (soft agar colony assay). Our results have shown that the iMSC secretome does not contain detectable levels of telomerase protein or nucleic acids encoding it, as well as it does not change the expression of pro-oncogenes and anti-oncogenes in primary fibroblasts and does not cause their malignization.

The data obtained are consistent with previously published studies investigating the pro-regenerative activity of the secretome of immortalized cells, since they do not declare any compromising properties of such secretomes [64,71]; however, all of them lack a comprehensive assessment of possible transforming (pro-malignant) activity of the secretome of immortalized cells.

It is worth noting that there are significant differences between immortalized and transformed cell cultures. Thus, immortalized cell cultures unlike transformed ones are characterized by genetic and phenotypic stability, retaining control over DNA integrity and cell cycle, sensitivity to contact inhibition and presence of growth factors in the culture medium [72,73,74].

Our study is a pioneering one in this field, and it is not possible to cover fully all the characteristics of such a complex object as the secretome of immortalized MSCs within the scope of a single work. Given the high potential for clinical translation of this technology, additional research is needed on the immunological, toxicological, and other safety aspects of the iMSC secretome, including the use of in vivo models not included in this study. As part of further safety studies of the iMSC secretome, in order to exclude the presence of the telomerase catalytic subunit in the secretome or to control its concentration below the maximum permissible one, it is necessary to use more sensitive and specific methods than Western blotting; in particular, those that allow to assess its catalytic activity, e.g., the Telomeric Repeat Amplification Protocol (TRAP).

It should be acknowledged that a number of tests we conducted (comparative analysis of protein and mRNA expression by pMSC and iMSC cultures, proteomic and transcriptomic analyses) require more repetitions in order to reliably assert that (1) there are no significant differences between the composition of secretomes, not only in terms of individual molecules, but across a wide range of proteins; and that (2) the secretomes of pMSCs and iMSCs do not alter the expression of pro- and anti-oncogenes in target cell cultures. However, the concordance of data obtained from a wide range of biochemical and physiological tests allows us to conclude that the secretome of immortalized MSCs exerts pronounced biological potency, lacks noticeable transforming activity, the content of key pro-regenerative molecules and their biological activity remain relatively stable during the prolonged passaging, and can be produced in clinically significant amounts. All this allows us to consider the iMSC secretome as a promising platform for the creation of a wide range of drugs for regenerative medicine.

## 4. Materials and Methods

### 4.1. Cell Cultures

The following cell cultures were used: HEK293T (ATCC, #CRL-3216™, Manassas, VA, USA) for lentiviral particle assembly; primary isolated human mesenchymal stromal cells (MSCs) for secretome production and immortalization; primary isolated human dermal fibroblasts for assessing of potential transforming (tumorigenic) activity of the secretome of iMSC (soft agar colony formation assay, transcriptome analysis).

Primary MSCs and human dermal fibroblasts were obtained from the Biobank of the Institute for Regenerative Medicine, Lomonosov Moscow State University (https://human.depo.msu.ru, accessed on 21 September 2025). All procedures with patient tissue samples were performed in accordance with the Declaration of Helsinki and approved by the Ethical Committee of Lomonosov Moscow State University (#IRB00010587), protocol #4 (2018).

HEK293T and the primary isolated human dermal fibroblasts were cultured in Dulbecco’s Modified Eagle’s Medium (DMEM), containing high glucose (Gibco, #11965092, Waltham, MA, USA), supplemented with 10% fetal bovine serum (FBS) (ThermoFisher Scientific, #26140079, Waltham, MA, USA), 1x antibiotic/antimycotic mixture (Gibco, #15240062) and 1x L-Gln (Gibco, #11539876). Similar culture medium with 2% inactivated FBS was used for the production of lentiviral particles.

The primary human MSCs were cultured in Advance Stem Cell Basal Medium (HyClone, #SH30879.02, Logan, UT, USA) supplemented with 10% Advance Stem Cell Growth Supplement (HyClone, #SH30878.01) and an antibiotic/antimycotic mixture (Gibco, #15240062). The obtained MSCs were characterized as plastic-adherent cells, expressing CD73, CD90 and CD105, lacking the expression of hematopoietic and endothelial markers CD14, CD19, CD20, CD34, CD45 and HLA-DR, and capable of adipogenic, osteogenic and chondrogenic differentiation in vitro, meeting the criteria established by the International Society for Cell Therapy (ISCT) [75,76]. All experiments with primary isolated MSCs (pMSCs) were performed up to 9 passages, while with the immortalized cultures (iMSCs)—up to 44 passages. All cell cultures mentioned hereinabove were cultured at 37 °C and 5% CO_2_, with culture medium change every 3–4 days.

Leydig cells were incubated in DMEM:F12 (Gibco, #11320033) supplemented with penicillin/streptomycin, GlutaMAX™ supplement (Gibco, #35050061), Sodium Pyruvate (100 mM) (Gibco, #11360070), 2% FBS, and Insulin-Transferrin-Selenium (ITS, PanEco, #F065, Moscow, Russia). Leydig cells were cultured at 35 °C and 5% CO_2_.

### 4.2. Animals

Eight-month-old Wistar rats (weighing 400–500 g) were used for isolating Leydig cells. Three-month old CD1 mice (weighing 21–24 g) were used as a source of sensory DRG. Animal housing and research procedures were conducted in compliance with Directive 2010/63/EU and in accordance with the local ethics committee (#90-G).

### 4.3. Immortalization of Human MSC Culture

To stabilize the qualitative and quantitative composition of the MSC secretome, the culture of primary isolated human MSCs was immortalized via the overexpression of human *TERT*, using the genetic construct pVLT-EF1a-hTERT-puroR as it was described previously [66,77]. Seven days after lentiviral transduction, the selection procedure of transduced MSC cultures was initiated by adding puromycin to cell culture medium at concentration of 1 μg/mL. The selection procedure was continued for 10–14 days, until complete cell death in the control culture of non-transduced MSCs was confirmed.

### 4.4. Obtaining of Secretome of pMSCs and iMSCs

The secretome of human MSC cultures was obtained as it was described previously [78,79]. Subconfluent MSC cultures (pMSCs at passages 2–9, iMSCs at passages 5–44) were thoroughly washed with Hanks’ solution (PanEco, #P020P) and then cultured for 7 days in serum-free DMEM Low Glucose (Gibco, #31885049), supplemented with the GlutaMAX™ supplement (Gibco, #35050061), Sodium Pyruvate (100 mM) (Gibco, #11360070) and 100 U/mL penicillin-streptomycin (Gibco, #15240062). To obtain extracellular vesicles (MSC-EV) fractions, the MSCs were incubated in the same conditions for 48 h.

Not significant MSC death was observed during conditioning in serum-free medium for 7 days. A detailed justification of the 7-day MSC conditioning protocol, with testing the MSC culture viability and analysis of the content of individual factors using ELISA, can be found in our previously published article [78].

The cell culture medium obtained was collected and centrifuged for 10 min at 300× *g* to remove cellular debris, and then concentrated 10-fold using ultrafiltration cartridges with a molecular weight cutoff (MWCO) of 10 kDa (Merck, Darmstadt, Germany). For experiments assessing the biological activity of the MSC secretome, serum-free DMEM Low Glucose, supplemented with GlutaMAX™, pyruvate and penicillin-streptomycin was processed in the same way to be used as a negative control (DMEM).

The secretome of MSC populations of close passages were grouped and analyzed together, allowing us to study the properties of the MSC secretome at early (p2–9), moderate (p12–18), late (p23–33) and very late (p41–44) passages.

### 4.5. MSC-EV Isolation, Characterization, Nanoparticle Tracking Analysis

To obtain the fraction of the secretome enriched with EV, the MSC culture medium was collected and centrifuged for 10 min at 300× *g* to remove cellular debris, and then concentrated 10-fold using ultrafiltration cartridges with a molecular weight cutoff (MWCO) of 1000 kDa (Merck, Darmstadt, Germany) as recommended in the Section 4 of the MISEV2023 [80]. The isolated EV of human adipose tissue-derived MSCs were previously characterized by several methods according to MISEV2023 recommendations and described in our recent papers [55,79].

In this study, the MSC-EV obtained have not been fully characterized, as the focus of our research was different. Here, we rely on the previously published results of our colleagues [58,59,60], who thoroughly compared the composition and properties of EV obtained from primary and TERT-immortalized MSC cultures. They demonstrated high similarity in size, composition, and properties of these two sets of MSC-EV. The results of these studies established that TERT-immortalization does not critically affect the composition and properties of EV produced by MSCs. In this study, we obtained EV using a well-established and previously published protocol [55,79].

To determine the particle size in the MSC secretome fraction enriched with EV we used the nanoparticle trajectory analysis (NTA) method on a ZetaView device (Particle Metrix, Inning am Ammersee, Germany). The measurement was performed in the “side light scattering” mode at 25 °C. A laser with a wavelength of 488 nm, a 10× lens, and a highly sensitive CMOS-type camera were used for the measurement. All measurements were performed in accordance with the recommendations of ASTM E2834-12 [81], using a standard operating procedure (SOP) optimized for EV analysis using the specific configuration (SOP “EV size, 488”) of the ZetaView device (Particle Metrix, Inning am Ammersee, Germany). Each sample was measured in 3–6 replicates to achieve a total of at least 2000 measured trajectories. The ZetaView Analyzer application (Particle Metrix, Inning am Ammersee, Germany) was used to process the results and construct graphs.

For experiments assessing the biological activity of MSC-EV, serum-free DMEM Low Glucose, supplemented with GlutaMAX™, pyruvate and penicillin-streptomycin was processed in the same way to be used as a negative control (DMEM).

### 4.6. Analysis of Qualitative and Quantitative Composition of Secretome of iMSC

To evaluate the qualitative and quantitative changes in the composition of the MSC secretome caused by the immortalization procedure, we performed (1) a proteomic analysis of the secretome of pMSCs and iMSCs, (2) an Enzyme-Linked ImmunoSorbent Assay (ELISA) of key neurotrophic and angiogenic factors (BDNF, uPA, VEGF and HGF) in the MSC secretome, (3) ELISA of the SASP (senescence associated secretory phenotype) components (MCP-1, PAI-1, IL-6), (4) qPCR to assess the changes in the mRNA content of key growth factors and cytokines in lysates of pMSCs and iMSCs.

For proteomic analysis, pMSCs and iMSCs were cultured until 80% confluency was reached (five to seven 100 mm cell culture dishes per sample), then deprived for 16 h in serum-free DMEM Low Glucose without phenol red (Gibco, #11054020). This was followed by conditioning in the similar medium for 24 h. Such short-term conditioning was used in order to reduce contamination of the secretome samples with MSC degradation products, which inevitably appear in the secretome during prolonged conditioning. The optimal parameters of cell culture conditioning for proteomic analysis were determined and described in detail in previously published articles [78,79,82]. The concentrations of therapeutic proteins (growth factors) obtained with this conditioning are extremely low and require a 100-fold or greater concentration of the secretome for their detection or for revealing a noticeable therapeutic effect. For this reason, conditioning for 24 h was used only for sample preparation for proteomic analysis, but not for in vitro/in vivo tests in this and other studies. The conditioned medium obtained was then concentrated 100-fold using MWCO 10 kDa cartridges.

Secretome concentrate containing 10–30 μg/μL of total protein was diluted 100-fold by 50 mM bicarbonate-ammonium buffer, pH 8.5, supplemented with 0.05% RapiGest reagent (Waters, #186001861, Milford, MA, USA). Then it was sequentially incubated with 2 mM tris-carboxyethylphosphine (Sigma-Aldrich, #C4706-10G, St. Louis, MO, USA) (one hour at 60 °C), with 4 mM methylmethanethiosulfonate (Sigma-Aldrich, #64306) (15 min, room temperature) and with 4 ng/μL trypsin (PanEco, #9002-07-7) (16 h at 37 °C). Afterwards, formic acid (Applichem, #131030, Darmstadt, Germany) was added to the samples to the final concentration of 0.1% and the resulting peptides were purified using HLB Oasis columns (Waters, #WAT106202). Chromatography-mass spectrometry analysis of the peptides was performed using the UltiMate 3000 RSLCnano UHPLC (ThermoFisher Scientific, Waltham, MA, USA) and a timsTOF Pro (Bruker, Billerica, MA, USA). The results obtained were analyzed using MaxQuant v2.6.7.0 (Max Planck Institute of biochemistry, Martinsried, Germany) according to the developers’ recommendations.

The content of BDNF, uPA, VEGF and HGF, which play a key role in the processes of vascularization and reinnervation of regenerating tissue, in the secretome was determined using Human Free BDNF Quantikine ELISA Kit (R&D, #DBD00, Minneapolis, MN, USA), Human uPA ELISA Kit (URK) (Abcam, #ab119611, Cambridge, MA, USA), Human VEGF Quantikine ELISA Kit (R&D, #DVE00) and Human HGF Quantikine ELISA Kit (R&D, #DHG00), respectively.

The content of SASP components (MCP-1, PAI-1 and IL-6) in the secretome of primary isolated MSC and iMSC was determined using: Human CCL2/MCP-1 ELISA Kit (R&D Systems, #DCP00), Human Serpin E1/PAI-1 Immunoassay (R&D Systems, #DSE100), Human IL-6 Quantikine ELISA Kit (R&D Systems, #D6050).

The mRNA content of key growth factors and cytokines in lysates of pMSCs and iMSCs was assessed using qPCR.

### 4.7. RNA Isolation, Reverse Transcription, qPCR

Total RNA was isolated from cell lysates using the Direct-zol RNA Miniprep kit (Zymo Research, Irvine, CA, USA, #R2052) according to the manufacturer’s instructions. Coding DNA (cDNA) synthesis was performed using the MMLV RT kit (Evrogen, Moscow, Russia, #SK021) according to the manufacturer’s recommendations. qPCR was performed using a 5X qPCRmix-HS SYBR mix (Eurogen, Moscow, Russia, #PK147L) according to the manufacturer’s instructions. The primer sequences and amplification parameters are listed in Appendix A.

### 4.8. Assessing the Pro-Regenerative Activity of the Secretome of iMSC on Potency Assays In Vitro

To assess the pro-regenerative potential of the secretome of iMSC cultures we used three diverse previously described in vitro models, able to determine the therapeutical potential of biological compositions. They are: (1) testosterone production by Leydig cells [52,53]; (2) TGF-b induced fibroblast-to-myofibroblast differentiation [54,55]; (3) mouse sensory ganglion neurite growth [56,57]. The Leydig cells-based and fibro-blast-to-myofibroblast differentiation models were previously validated for assessing the pharmacological activity of biological compositions for regenerative medicine to treat fibrosis and male infertility, respectively [52,53,54,55]. The Leydig cells-based model is used for MSC secretome quality control within the clinical trials being conducted [No. NCT06869863, https://clinicaltrials.gov/study/NCT06869863, accessed on 21 September 2025].

Although these models are cellular, they reflect the biological activity of drugs (they are potency assays), in particular, the secretome of MSCs, as pro-regenerative drugs in in vivo models [52,53,54,55,56,57,78].

#### 4.8.1. Stimulation of Testosterone Production by Leydig Cell Culture (Leydig Cell Culture-Based Potency Assay)

The Leydig cell culture-based model for evaluating the biological activity of MSC secretome is built on the ability of the MSC secretome to stimulate the production of testosterone by Leydig cells. This model has been described in detail previously [52], including an assessment of dose dependence and determination of potency units (a change in the secretory activity of Leydig cells of ≥15% indicates the biological activity of the secretome). The validation data of this model served as the basis for registering the patent “Assay for evaluating the secretory activity of model cells” #WO2024177532A1 (https://patents.google.com/patent/WO2024177532A1/en, accessed on 21 September 2025).

Isolation of Leydig cells was performed as it was previously described [52]. Briefly, rats were anesthetized in the CO_2_ atmosphere and then euthanized by cervical dislocation. Rat testes were isolated, washed in Hanks’ Balanced Salt Solution (HBSS) (PanEco, Moscow, Russia) supplemented with 5% antibiotic penicillin/streptomycin (Gibco, ThermoFisher Scientific, Waltham, MA, USA), decapsulated, washed and incubated in serum-free DMEM Low Glucose supplemented with 2.5 mg/mL trypsin and 10 ug/mL DNAse for 4–10 min at 34–35 °C with shaking. This caused the interstitial cells to dissociate from the seminiferous tubules. To inactivate the enzymes, complete growth medium was added. The cell suspension was gently mixed by inverting the tube and left for 5 min to allow the tubules to settle down. The supernatant fraction containing interstitial cells was moved into a new tube and the procedure of mixing and supernatant removal was repeated again (in total, five times). The obtained cell suspension was centrifuged at 300× *g* and the residue was resuspended in complete growth medium for Leydig cells. The Leydig cell suspension was filtered through a Falcon^®^ 100 µm Cell Strainer (Corning, Corning, NY, USA, #352360) and seeded in 48-well culture plates.

Previously, we have described an in vitro Leydig cell culture-based potency assay for assessing the MSC secretome activity for treating idiopathic male infertility [52]. To assess how MSC immortalization affected the activity of MSC secretome, we used this previously described model.

For this, the day after seeding (see hereinabove) Leydig cell cultures (in some wells) were washed three times with 500 µL of HBSS and conditioned in serum-free DMEM Low Glucose without phenol red, supplemented with penicillin/streptomycin, Glutamax-I, and pyruvate. The next day, the conditioned medium was collected and centrifuged at 300× *g* for 10 min. The supernatant (Control 2 day—positive control) was stored at −80 °C. On the same day, Leydig cell cultures in the remaining wells were washed three times with HBSS and the medium was changed to either the pMSC or iMSC secretomes (experimental groups), or serum-free DMEM Low Glucose without phenol red, supplemented with penicillin/streptomycin, Glutamax-I, and pyruvate (control 4 day—negative control). Two days later, the Leydig cell secretome was collected and centrifuged at 300× *g* for 10 min. The supernatant was stored at −80 °C. The model is considered working if the testosterone concentration in “control 2 day” is significantly higher than in “control 4 day”. This reflects the normal behavior of Leydig cells in culture, as their ability to secrete testosterone decreases over time.

The concentration of testosterone in Leydig cell secretome was determined using DBC Testosterone ELISA CAN-TE-250 (#CAN-TE-250, Diagnostics Biochem, London, ON, Canada).

#### 4.8.2. Prevention of Fibroblast-to-Myofibroblast Differentiation (In Vitro Model of Fibrosis)

To induce fibroblast-to-myofibroblast differentiation, fibroblasts were seeded in cell culture plates at a concentration of 1.5 × 10^4^ cells per cm^2^, grown for 1 day, and serum-deprived overnight in serum-free DMEM Low Glucose. Then, the fresh DMEM with 5-ng/mL TGFb (R&D Systems, Minneapolis, MN, USA) with or without MSC-EV was applied together. The cells cultured in DMEM Low Glucose were used as negative control, whereas those cultured in DMEM Low Glucose with 5 ng/mL TGFb were used as positive control. Plates were incubated in a CO_2_-incubator at 37 °C and analyzed after 4 days.

To analyze myofibroblast differentiation, the cells were fixed with 4% paraformaldehyde in Phosphate-Buffered Saline (PBS) (#P061, PanEco, Moscow, Russia) for 10 min at room temperature (RT) and permeabilized with 0.1% Triton X-100 for 10 min. After the blocking of nonspecific binding with 10% normal goat serum on 1% bovine serum albumin (BSA) (#PM-T1725.50, PanEco, Moscow, Russia), the cells were incubated with antibody to a-SMA (Abcam, #ab32575, Cambridge, MA, USA) or rabbit IgG (Santa Cruz Biotechnology, #NSC-2025, Dallas, TX, USA) overnight at 4 °C. Antibody detection was performed using secondary antibodies conjugated with Alexa Fluor 488 (Invitrogen, #A11037, Waltham, MA, USA) for 1 h at RT in the dark. The nuclei were counterstained with 4′,6-Diamidino-2-Phenylindole (DAPI) (Sigma-Aldrich, #D9542, St. Louis, MO, USA). Images were obtained using Leica DMi8 with Leica DFC 7000 T camera (Leica Microsystems GmbH, Wetzlar, Germany) and processed using the LasX (Leica Microsystems GmbH) and FIJI ImageJ 1.54 p software (GitHub Inc., San Francisco, CA, USA). This experiment was repeated 3 times.

#### 4.8.3. Stimulation of Neurite Growth of Murine Sensory Ganglions (In Vitro Model of Neuritogenesis)

To evaluate the ability of pMSC and iMSC secretomes to stimulate neurite growth (as an important stage of tissue regeneration), we used an in vitro murine DRG explant model described previously [56,57]. Mice were euthanized by slowly increasing the concentration of carbon dioxide in the inhaled air. The DRG located paravertebrally T8-L5 were isolated, cleaned of the connective tissue and seeded in a drop of Corning^®^ Matrigel^®^ Growth Factor Reduced (GFR) Basement Membrane Matrix (Corning, #354230). Each group included at least 5 ganglions. The obtained In Vitro cultures were cultivated in serum-free DMEM/F12 medium containing 1% L-glutamine and 1% antibiotic/antimycotic mixture supplemented with 10% of 50-fold concentrated pMSC or iMSC secretome. Serum-free DMEM/F12 growth medium with 1% of L-glutamine and 1% of antibiotic/antimycotic mixture was used as a negative control. The dynamics of neurite growth in the obtained ganglion cultures were recorded every 3 days (up to 20 days) by taking pictures using an inverted wide-field microscope, the Nikon Eclipse Ti2, equipped with a Kinetix digital monochrome camera (Teledyne Photometrics, Seattle, WA, USA) and a Ri2 color camera (Nikon, Tokyo, Japan). The images obtained were analyzed using NIS Elements AR 5.40.02 and FIJI ImageJ 1.54 p software (GitHub Inc.). The identity of the observed microscopical structures with nerve fibers was confirmed in some samples by immunohistochemical staining with antibodies to the neuron-specific heavy neurofilament protein NF200 (Merck, #5389) as previously described [83,84].

### 4.9. Study of the Potential Transforming Activity of Secretome of iMSC

To study the potential transforming activity of the iMSC secretome, we evaluated: (1) the presence of telomerase in the iMSC secretome as a protein or telomerase-encoding nucleic acid (DNA, RNA); (2) the ability of the iMSC secretome to initiate colony growth in a culture of primary isolated dermal fibroblasts in a soft agar colony formation assay; (3) the transcriptomes of primary isolated dermal fibroblast cultures (the expression changes in oncogenes and anti-oncogenes, above all) treated with the secretome of pMSCs or iMSCs.

#### 4.9.1. Detection of Telomerase Protein and Nucleic Acids Encoding It in the Secretome

To detect the telomerase or nucleic acids encoding it in the secretome of MSCs, the secretome was obtained as described above, and concentrated 100-fold using MWCO 10 kDa cartridges. Total DNA from the concentrated secretome was isolated using phenol-chloroform extraction. Total RNA isolation from the concentrated MSC secretome, cDNA synthesis, and real-time PCR were performed as described above. The primer sequences and amplification parameters are listed in Appendix A.

Telomerase protein detection in the concentrated secretome was performed using polyacrylamide gel (PAAG) electrophoresis, followed by protein transfer to PVDF membrane and staining with the Anti-Telomerase reverse transcriptase antibody [2C4] (Abcam, Cambridge, MA, USA, #ab5181). Polyacrylamide gell pocket protein loading normalization (9 ug of total protein per well) was based on the total protein concentration of each sample determined using the Pierce™ BCA Protein Assay Kit (ThermoFisher Scientific, Waltham, MA, USA, #23227), following the manufacturer’s recommendations. To verify the activity of the Anti-Telomerase reverse transcriptase antibody [2C4] (Abcam, Cambridge, MA, USA, #ab5181), lysates of immortalized iMSCs were used as positive control.

To demonstrate the sensitivity of the PAAG-electrophoresis + Western Blotting (WB) method for protein detection in MSC secretomes, we stained the obtained PVDF membranes for alpha Tubulin, the intracellular cytoskeleton protein whose concentration is extremely low in cell secretome. For alpha Tubulin detection, the PVDF membranes were stained with alpha Tubulin Antibody (B-5-1-2) (Santa Cruz, Dallas, TX, USA, #sc-23948). P-RAM Iss (Imtek, Moscow, Russia) were used as secondary antibodies. To establish the approximate molecular weight of the detected proteins we used the PageRuler™ Prestained Protein Ladder molecular weight marker, 10 to 180 kDa (ThermoFisher Scientific, #26616). The PVDF membrane was stained with the Clarity™ Western ECL Substrate kit (Bio-Rad, Hercules, CA, USA, #1705060) and the signal was detected using the Bio Rad ChemiDoc MP Imaging System (Bio-Rad, Hercules, CA, USA) for up to 300 s. This experiment was repeated 4 times.

As part of further safety studies of the iMSC secretome, in order to keep the telomerase catalytic subunit concentration below the maximum permissible level, it is necessary to use more sensitive and specific methods than Western blotting, in particular, those that make it possible to assess its catalytic activity, e.g., the Telomeric Repeat Amplification Protocol (TRAP) [85].

#### 4.9.2. Soft Agar Colony Formation Assay

To evaluate the potential transforming activity of the iMSC secretome, we used the previously described soft agar colony formation assay, which was based on the ability of transformed cells to form colonies in agar [86]. For this test, we used human dermal fibroblasts (HDF) of 5–8 passages obtained from two different donors.

The wells of a 12-well plate were pre-coated with 1% agarose solution (Panreac, #A2114,0250, Barcelona, Spain) based on serum- and antibiotic-free DMEM, sterilized by autoclaving. Fibroblasts were detached from cell culture dishes, resuspended in complete growth medium, filtered through Falcon^®^ 40 µm Cell Strainer (Corning, Corning, NY, USA, #352340) to remove cell aggregates, and counted. The autoclaved sterilized solution of 0.8% agarose on serum- and antibiotic-free DMEM was cooled to 39 °C, mixed with an equivalent volume of DMEM, supplemented with 2x L-Glutamine solution (Gibco, #11539876), 2x antibiotic/antimycotic mixture solution (Gibco, #15240062) and 10% bovine serum albumin, preliminary sterilized by filtering through a 0.22 µm Millex-GP Syringe Filter Unit (Merck, #SLGPR33RB) and warmed up to 39 °C. The prepared suspension of human dermal fibroblasts was added to the resulting mixture of agarose solution and DMEM to achieve the final concentration of 1000 cells/mL. The resulting cell suspension was gently mixed and layered by 500 µL into the wells of the pre-prepared 12-well plate.

The plates were maintained under sterile conditions at room temperature for at least 40 min to allow agarose solidification. Subsequently, 500 µL of the test solutions were gently layered onto the agarose surface. These solutions comprised 10-fold concentrated secretomes from primary MSCs (pMSC) and immortalized MSCs (iMSC), fibroblast growth medium serving as a negative control, and positive controls consisting of 0.01% dimethyl sulfate (DMS) or 3 mM sodium nitrite (NaNO2) diluted in complete growth medium [87,88]. Each group was performed in triplicates for both HDF donors. Test solutions were refreshed every three days over a two-week period. In the DMS-treated group, 6 h after the start of the experiment, the DMS solution was replaced with the complete fibroblast growth medium, which was further used in this group, since exposure of cells to the 0.01% DMS solution for 24 h resulted in the death of the entire cell culture.

Photo registration of cell cultures in transmitted light was performed immediately after the first addition of test solutions (0-day point) to ascertain the absence of cell aggregates at the beginning of the experiment, and 14 days since the beginning of the experiment, using Nikon Eclipse Ti inverted microscope (Nikon, Tokyo, Japan) equipped with a camera. The number of single cells, cell duplets and multicellular structures in each test group was taken into account. The threshold for exerting the potential transformative activity was considered to be the appearance of at least one multicellular colony in the study group. The researcher who counted the colonies and processed the raw data was blinded.

#### 4.9.3. Transcriptomic Analysis of Primary Human Fibroblasts Treated with the iMSC Secretome

To assess the potential ability of the iMSC secretome to alter the expression of pro- and anti-oncogenes in target cells, we analyzed the transcriptome of primary human dermal fibroblasts treated with the secretome of pMSCs and iMSCs. For this purpose, we used cultures of primary human dermal fibroblasts of 5–8 passages of 80–90% confluent. In experimental groups, the growth medium was replaced with the non-concentrated secretome of pMSCs or iMSCs. In the negative control group, cells were cultured in complete fibroblast growth medium, and in the positive control groups, mutagen solutions (3 mM sodium nitrite or 0.01% DMS in complete fibroblast growth medium) were used [87,88]. In the DMS group, 6 h after the start of the experiment, the DMS solution was replaced with the complete fibroblast growth medium. Test solutions were renewed every 3 days, and 7 days after the start of the experiment, the fibroblast cultures were lysed for total RNA isolation as described above. About 800 ng of total RNA (for each group) was utilized for library construction employing the NEBNext^®^ Poly(A) mRNA Magnetic Isolation Module alongside the NEBNext^®^ Ultra II™ Directional RNA Library Prep Kit for Illumina (New England Biolabs, Ipswich, MA, USA), following the manufacturer’s guidelines.

Libraries’ integrity and quality were evaluated using the Bioanalyzer 2100 (Agilent, Santa Clara, CA, USA), and confirmed by quantitative PCR. Sequencing was conducted on an Illumina NovaSeq 6000 platform with 61 bp paired-end reads. Subsequent data processing and analysis were performed using the STAR aligner v.2.7.11b (Github Inc.), Subread v.2.1.1 (Ohio Supercomputer Center, Columbus, OH, USA), and DESeq2 v.3.21 (Bioconductor, Boston, MA, USA) software tools. The resulting gene lists were annotated using the g:Profiler resource (https://biit.cs.ut.ee/gprofiler/, accessed on 19 May 2025).

### 4.10. Statistical Analysis

Statistical analysis was performed using the SigmaPlot11.0 software (Systat Software, Inc., Erkrath, Germany). Numerical data were assessed for normality of distribution using the Kolmogorov–Smirnov criterion. Differences between experimental and control groups were analyzed using Student’s *t*-test (for pairwise comparisons) or the analysis of variance (ANOVA) Holm–Sidak test (for multiple comparisons) if the distribution was normal. ANOVA on ranks (Dunn’s test) was used when the distribution was not normal. Data are presented as mean ± standard deviation or median (25%; 75%) depending on the test used. Differences between groups were considered significant at *p* < 0.05.

## 5. Conclusions

MSC immortalization makes it possible to overcome some limitations of primary MSCs cultures as cell producers. The immortalization of an MSC culture allows stabilizing the qualitative and quantitative composition of its secretome for more than 30 passages. The secretome of iMSCs, even at the 30th passage, retains its ability to stimulate regeneration with efficiency equal to early passages of primary MSCs. The iMSC secretome does not contain detectable amounts of telomerase and does not possess any transforming activity. Given the high potential for clinical translation of this technology, additional research is needed on the immunological, toxicological, and other safety aspects of the iMSC secretome, including the use of in vivo models not included in this study. However, the concordance of data obtained from a wide range of biochemical and physiological tests allows us to conclude that the secretome of immortalized MSCs exerts pronounced biological potency, lacks noticeable transforming activity, the content of key pro-regenerative molecules and their biological activity remains relatively stable during the prolonged passaging, and can be produced in clinically significant amounts. All this allows us to consider the iMSC secretome as a promising platform for the creation of a wide range of drugs for regenerative medicine.

## Figures and Tables

**Figure 1 ijms-26-09322-f001:**
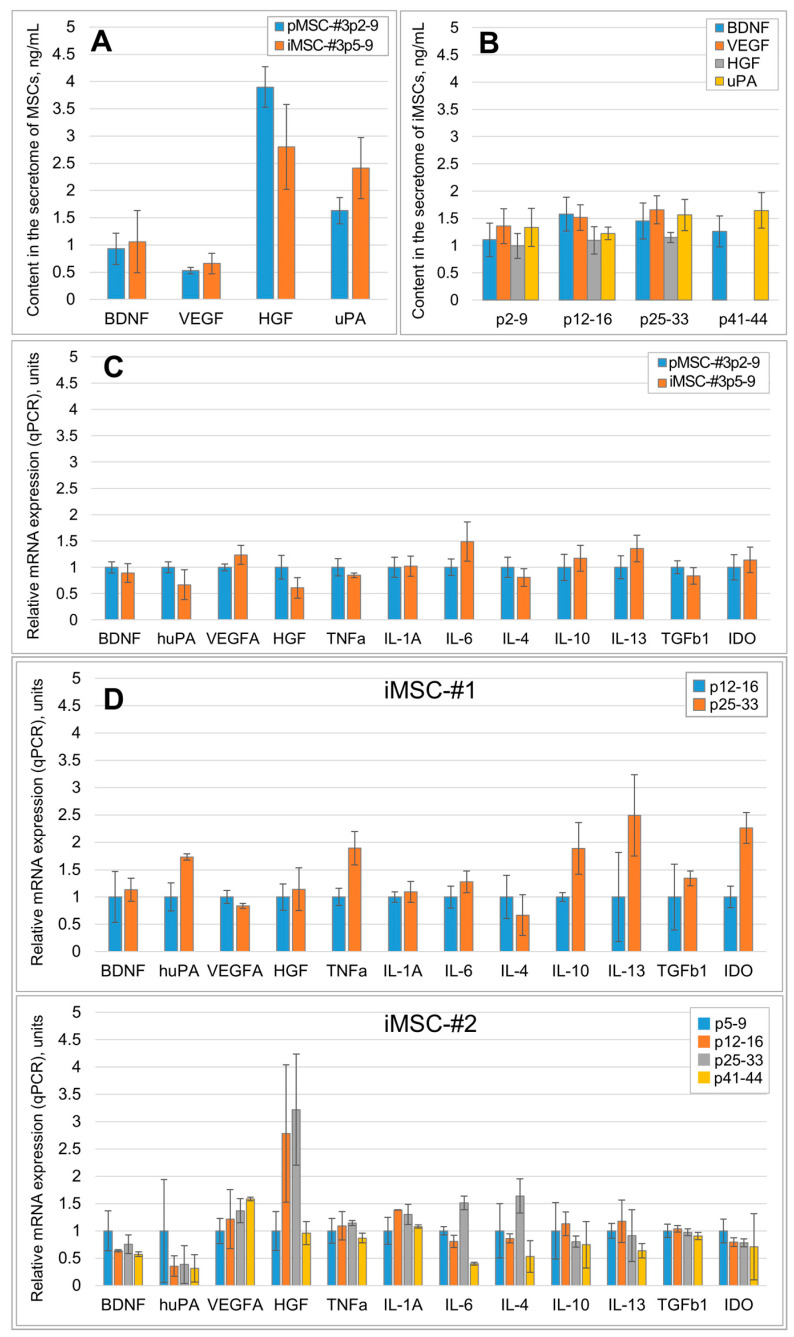
Immortalization stabilizes the qualitative and quantitative composition of the MSC secretome. (**A**) The effect of MSC culture immortalization on the production of major neuroprotective and angiogenic factors. pMSC-#3p2-9—primary MSCs sample 3 passages 2–9, iMSC-#3p5-9—immortalized MSCs sample 3 passages 5–9 (ELISA), n ≥ 3; (**B**) Production of major neuroprotective and angiogenic factors by iMSC-#2 culture in dynamics (ELISA). p5-9—passages 5–9, etc., n = 3. (**C**) Comparison of mRNA expression (qPCR) of major neuroprotective and pro-angiogenic factors, pro-inflammatory and anti-inflammatory molecules in MSC cultures before immortalization (pMSC-#3p2-9—primary MSCs sample 3 passages 2–9) and after it (iMSC-#3p5-9—iMSC sample 3 passages 5–9), etc., n = 3. (**D**) Relative mRNA expression (qPCR) of major neuroprotective and angiogenic factors, pro-inflammatory and anti-inflammatory molecules in iMSC-#1 and iMSC-#2 cultures in dynamics; p5-9—passages 5–9, etc., n = 3. Data are presented as mean ± standard deviation. No statistically significant differences have been detected between experimental and control groups.

**Figure 2 ijms-26-09322-f002:**
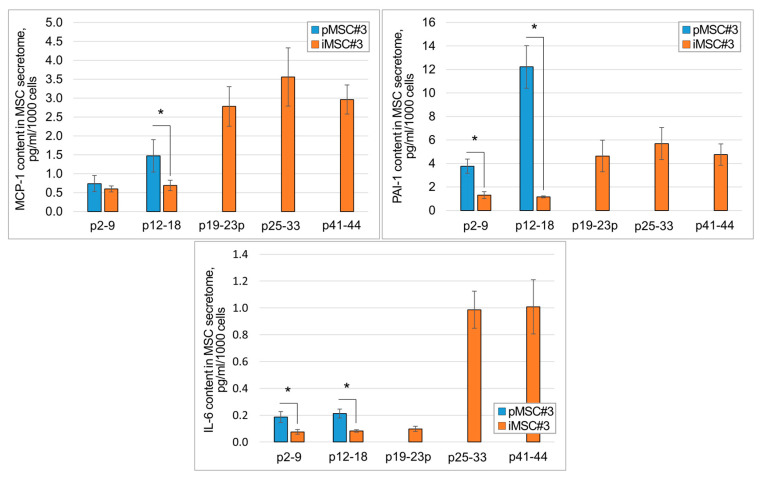
Comparative content of SASP components in the secretome of primary isolated and iMSC in dynamics; p2-9—passages 2–9, etc. Data are presented as mean ± standard deviation. *—*p* < 0.05, n ≥ 3, Student’s *t*-test.

**Figure 3 ijms-26-09322-f003:**
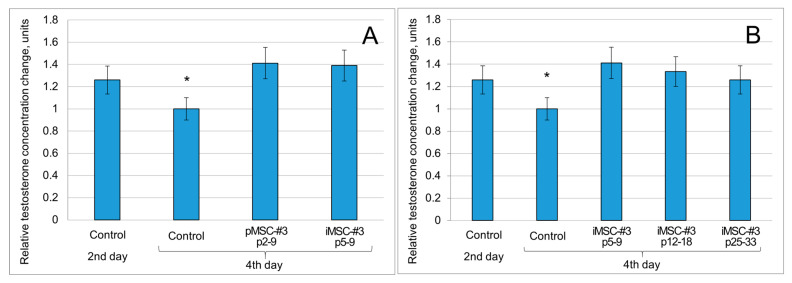
The iMSC secretome stimulates the secretory activity of Leydig cells. (**A**) MSC secretome-stimulated Leydig cells produce more testosterone compared to control cells (day four). The secretome of iMSCs (passages 5–9) stimulates testosterone production by Leydig cells as effectively as the secretome of pMSCs does. (**B**) The secretome of iMSCs retains its activity even for iMSCs at passages 12–18 and 25–33. Testosterone concentrations are normalized to that in the “control day four” sample; p2–9—passages 2–9, etc. Data are presented as mean ± standard deviation. *—*p* < 0.05 (compared to other groups), n ≥ 3, Student’s *t*-test.

**Figure 4 ijms-26-09322-f004:**
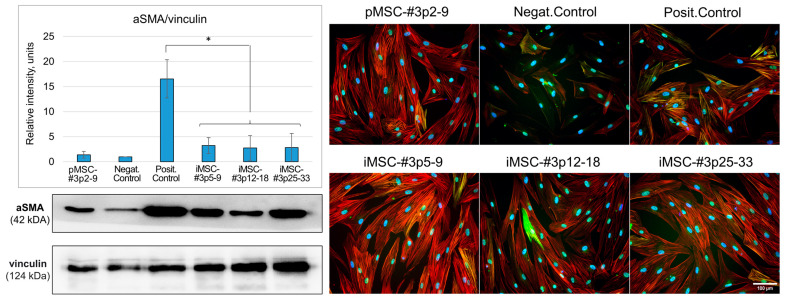
The iMSC secretome EV-enriched fractions prevent fibroblast-to-myofibroblast differentiation (an in vitro model of fibrosis). The **left panel**—results of Western blot analysis of cell lysates stained with antibodies to alpha smooth muscle actin (aSMA); the signal obtained was normalized to vinculin; n = 3. The **right panel** contains immunocytochemistry micrographs of fibroblast cultures treated secretomes of pMSCs and iMSCs. Red staining—phalloidin; green staining—aSMA; blue staining (4′,6-Diamidino-2-Phenylindole, DAPI)—nuclei. p2-9—passages 2–9, etc., (*—*p* < 0.05, n = 3, Student’s *t*-test).

**Figure 5 ijms-26-09322-f005:**
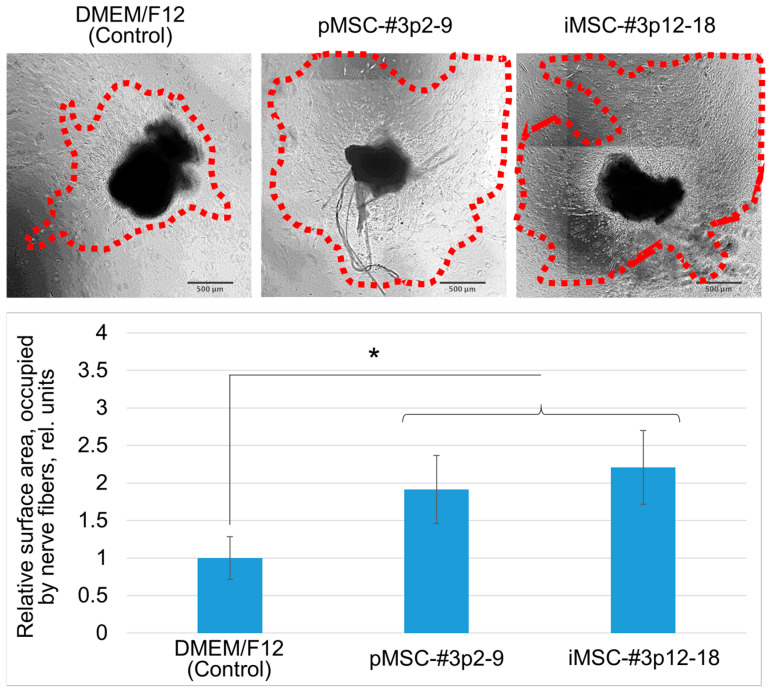
The iMSC secretome stimulates neurite outgrowth from the explants of murine sensory DGR (an in vitro model of neuritogenesis). The **upper panel**—samples of microphotographs of murine sensory DRG cultured in serum-free DMEM/F12 or secretomes of primary and immortalized MSCs, 20th day. The **lower panel**—results of analysis of the surface area, occupied by nerve fibers, grown from the ganglion (*—*p* < 0.05, n ≥ 5, ANOVA Holm–Sidak test). The red dotted line marks the growth front of the neurites.

**Figure 6 ijms-26-09322-f006:**
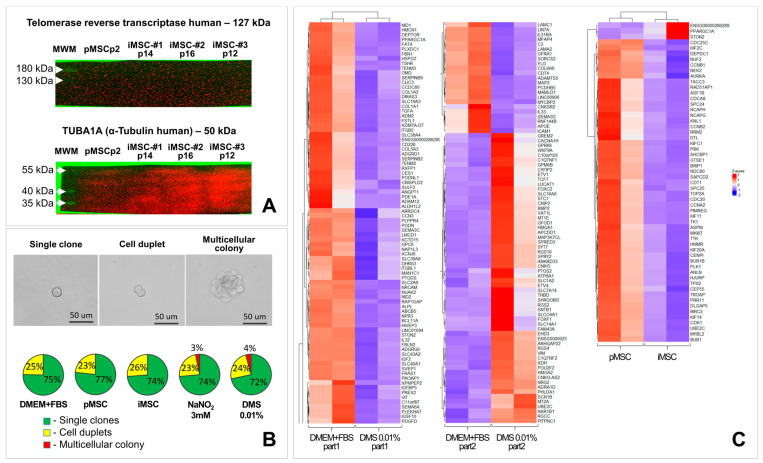
The secretome of immortalized MSCs does not contain detectable amounts of telomerase and reveals no transforming activity. (**A**) Western blot results of the 100-fold concentrated secretome samples of primary MSCs, passage 2 (pMSC-p2) and iMSC cultures #1–3 of passages 12–16 (iMSC-#1p14, iMSC-#2p16, iMSC-#3p12) stained for the human telomerase reverse transcriptase. No telomerase was detected in the samples. No statistical analyses were performed in this qualitative test. The representative results of four independent experiments are presented. (**B**) The iMSC secretome does not induce fibroblast colony formation in soft agar colony formation assay. Top panel—micrograph samples of a single cell, cell duplet and a multicellular colony. Bottom panel—the percentage of single cells, cell duplets and cell colonies in the studied groups on day 14th. No statistical analyses were performed, it is a qualitative test. The representative results of two independent experiments are presented, each of which was conducted in triplicates. (**C**) The iMSC secretome does not change the expression of pro- and anti-oncogenes in a culture of primary human dermal fibroblasts. DMEM+FBS (control)—complete fibroblast growth medium, pMSCs—primary MSCs, iMSCs—immortalized MSCs, DMS—dimethyl sulfate. No statistical analysis is possible due to the small sample size (n = 2).

## Data Availability

Data are available upon request.

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
