# Peer review of "Safety and Regenerative Properties of Immortalized Human Mesenchymal Stromal Cell Secretome"

_ijms, 2025, doi:10.3390/ijms26199322_

Round 1

Reviewer 1 Report

Comments and Suggestions for Authors

This manuscript explores the therapeutic potential of secretome derived from immortalized human mesenchymal stromal cells, assessing their content, variability during passaging, biological activity, and safety compared to primary MSC secretome. The paper addresses an important and novel topic of potential use of iMSC-derived secretome for clinical applications in regenerative medicine and molecular cell therapy. However, an entire dataset is performed in vitro, which is insufficient for clinical use. The manuscript is poorly structured, with low figure aesthetics that significantly deteriorates the quality of the experimental work.

Major points:

  1. While the performed experiments are interesting and novel, they do not support strong conclusions of this manuscript. The manuscript should be re-written to reflect that, while promising, further experimental and in vivo preclinical validations are required for justifying clinical application of this system.
  2. Did the Authors perform secretome analysis? They are citing 2 papers but do not present the data on the figure. This issue should be clearly pointed out. If the Authors use published data, how did they analyze it?
  3. Did the Authors perform exosome characterization, including size/concentration assessment, Western blot or flow cytometry for exosome vesicle markers?
  4. Immortalized cell-derived products raise concerns about immune response upon administration. Immune response activation was not tested.

Minor points:

  1. Introduction could be broader and explain in more detail the secretome of primary MSC and the application of secreted proteins in tissue regeneration. The final paragragraph explaining what has been done in this study should also be more detailed and with clearer hypothesis.
  2. Pay attention to abbreviations, when used the first time, full name should be used, afterwards, the acronym.
  3. Improve transition between paragraphs, especially when shifting from one experimental result to another.
  4. Statistical details (number of replicates, SD/SEM, p-values) should be presented on figure legends.

Comments on the Quality of English Language

A few English language corrections throughout the text.

Author Response

Dear Reviewer, Thank you very much for your time and valuable comments! We believe that the manuscript has become much better. Here below, we provide a point-by-point response to your comments. All changes to the original manuscript are highlighted in green.

This manuscript explores the therapeutic potential of secretome derived from immortalized human mesenchymal stromal cells, assessing their content, variability during passaging, biological activity, and safety compared to primary MSC secretome. The paper addresses an important and novel topic of potential use of iMSC-derived secretome for clinical applications in regenerative medicine and molecular cell therapy. However, an entire dataset is performed in vitro, which is insufficient for clinical use.

Yes, we agree with you that the data provided is insufficient for the clinical translation of the secretome of immortalized MSCs (iMSC), and we don’t assert this. Our study is one of the first in this field. The iMSC secretome seems to be a highly promising tool for regenerative medicine [doi 10.3390/ijms241612716], but it has not been sufficiently studied at this point. The investigation of the iMSC secretome properties requires a series of additional studies with a large number of replicates, using a vast number of models, including in vivo models, with an assessment of the immunomodulatory, potential toxic or carcinogenic properties of such secretome. We have supplemented our manuscript with these limitations in the last two paragraphs of the Discussion section.

The manuscript is poorly structured.

We have revised the text in accordance with your comments and recommendations: we have added connecting sentences, expanded the introduction, and formulated the hypothesis of the study more clearly. The illogicality of the presentation of the study is apparent and is most likely due to the structure of the manuscript in IJMS, where the results are presented before the materials and methods. To improve the logic, connecting sentences and reasoning have been added to the results section.

with low figure aesthetics that significantly deteriorates the quality of the experimental work.

We have revised our graphs and figures and tried to bring them into a common format.

Major points:

  1. While the performed experiments are interesting and novel, they do not support strong conclusions of this manuscript. The manuscript should be re-written to reflect that, while promising, further experimental and in vivo preclinical validations are required for justifying clinical application of this system.

Yes, we agree with you that the data provided is insufficient for the clinical translation of the secretome of immortalized MSCs (iMSC), and we don’t assert this. Our study is one of the first in this field. The iMSC secretome seems to be a highly promising tool for regenerative medicine [doi 10.3390/ijms241612716], but it has not been sufficiently studied at this point. The investigation of the iMSC secretome properties requires a series of additional studies with a large number of replicates, using a vast number of models, including in vivo models, with an assessment of the immunomodulatory, potential toxic or carcinogenic properties of such secretome. We have supplemented our manuscript with these limitations in the last two paragraphs of the Discussion section.

  1. Did the Authors perform secretome analysis? They are citing 2 papers but do not present the data on the figure. This issue should be clearly pointed out. If the Authors use published data, how did they analyze it?

The question is not clear. What analysis does the Reviewer mean? The secretomes of primary and immortalized MSCs were analyzed using various methods. If the Reviewer is asking about the analysis of the MSC secretome composition in previously published articles, then such analysis, depending on the passage and characteristics of the donor, was previously conducted in other studies (also conducted by our team) [doi: 10.1186/s13287-015-0209-8, 10.3390/ijms26157164, 10.1134/S1990750824601085, 10.5966/sctm.2013-0014, 10.3390/ijms25010290, 10.2217/rme.12.16]. When writing this publication, we relied on the results and conclusions obtained in previously published works, but did not analyze them further—more details about the analysis results can be found in the original studies. In this study, a sufficient number of original and previously unpublished results were obtained, which does not require supplementing it with the results of other third-party studies.

If the Reviewer means the phrase «packaging of microRNAs (including neuroprotective ones) into extracellular vesicles (HNRNPU) [48, 49]», the references 48 and 49 are related to the studies that describe nature and function of HNRNPU, not the results of our proteome analysis. Limited funding for this study did not allow for a large number of replicates (n = 1), which prevents us from performing statistical analysis and quantitatively assessing changes in the expression of a wide range of proteins. However, the data obtained allow us to conclude that the qualitative composition of the MSC secretome is preserved after immortalization. The raw data of the proteomic analysis were deposited in the PRIDE repository (https://www.ebi.ac.uk/pride/archive/projects/PXD067843) and information about this was added to the text of the manuscript. The quantitative analysis of expression of certain growth factors and cytokines was performed using ELISA and qPCR. This information was specified in the text of the manuscript and in the limitations of the study.

  1. Did the Authors perform exosome characterization, including size/concentration assessment, Western blot or flow cytometry for exosome vesicle markers?

Yes, we did not conduct a full characterization of the properties of extracellular vesicles (EVs) in this study, as the focus of our research was different. In this study, we relied on the results previously obtained by our colleagues [doi 10.3390/cells13100861, 10.1155/2020/1289380, 10.1101/2024.12.05.626964], who thoroughly compared the properties of EVs obtained from primary and TERT-immortalized MSC cultures, and showed their high similarity in size, composition, and properties. The results of these studies established that immortalization using TERT does not critically affect the composition and properties of EVs produced by MSCs. In this study, we obtained EVs using a well-established and previously published protocol [10.3390/cells9051272, 10.1038/s12276-023-01017-w]. Using NTA in this study, we confirmed that the secretomes of TERT-immortalized MSCs contain EVs that are similar in size and concentration to those of primary MSCs (Figure S1). These data were added to the materials and methods (section 4.5), results, and supplementary materials of the manuscript.

  1. Immortalized cell-derived products raise concerns about immune response upon administration. Immune response activation was not tested.

Yes, we agree with you that the data provided is insufficient for the clinical translation of the secretome of immortalized MSCs. Immunological, toxicological, and other safety aspects of MSC secretome require further thorough investigation, which is the reasoning for conducting additional studies. We have supplemented our manuscript with these limitations in the last two paragraphs of the Discussion section.

Minor points:

  1. Introduction could be broader and explain in more detail the secretome of primary MSC and the application of secreted proteins in tissue regeneration. The final paragragraph explaining what has been done in this study should also be more detailed and with clearer hypothesis.

The introduction was expanded and the hypothesis was formulated more clearly.

  1. Pay attention to abbreviations, when used the first time, full name should be used, afterwards, the acronym.

All abbreviations have been expanded; the necessary changes have been made to the text of the manuscript.

  1. Improve transition between paragraphs, especially when shifting from one experimental result to another.

To improve the logic, connecting sentences and reasoning were added to the results section.

  1. Statistical details (number of replicates, SD/SEM, p-values) should be presented on figure legends.

Statistical details, where they were missing, have been added. We have checked their presence under each figure. If no statistically significant differences between groups was detected, we also noted this.

Reviewer 2 Report

Comments and Suggestions for Authors

Karagyaur and colleagues assessed some biological effects and safety of secretomes produced by human mesenchymal stromal cells (hMSC) immortalized with telomerase (TERT). Proteomics analysis showed that TERT/hMSC secretomes exhibit only minor differences when compared with normal hMSC secretomes. Also, TERT/hMSC secretomes did not contain TERT DNA and did not stimulate fibroblast colony formation in soft agar, which demonstrates their safety.  They efficiently suppressed the TGFbeta-induced in vitro fibrosis and stimulated the production of testosterone in Leydig cell culture. The study shows a significant potential of TERT/hMSC secretomes as a safe anti-fibrosis agent and stimulator of spermatogenesis. The safe immortalization of MSC opens an opportunity to standardize and highly enhance the production of their secretomes, which could be used to repair damaged tissues.

Critiques:

  1. The term “regenerative properties” in the title of the present paper is an overkill. Experiments with 3D organoid cultures are needed to justify its use.
  2. Immunofluorescence images in Figure 4 should be presented without phalloidin staining, which hinders the observation of secretomes’ effect on alpha-SMA expression.
  3. In figure 5A, a positive Western control of TERT should be presented. Also, tubulin gel is of low quality.

Author Response

Dear Reviewer, Thank you very much for your time and valuable comments! We believe that the manuscript has become much better. Here below, we provide a point-by-point response to your comments. All changes to the original manuscript are highlighted in green.

Critiques:

  1. The term “regenerative properties” in the title of the present paper is an overkill. Experiments with 3D organoid cultures are needed to justify its use.

I beg to differ on that point. The biological models we use are validated and previously published models for assessing the pharmacological activity of drugs (one of which is already undergoing clinical trials, No. NCT06869863, https://clinicaltrials.gov/study/NCT06869863) in the field of regenerative medicine for pathologies such as fibrosis and male infertility [doi: 10.3390/ijms23169414, 10.3390/cells9051272, 10.1016/j.bbrc.2024.150574]. Although these models are cellular, they reflect the biological activity of drugs (they are potency assays), in particular, the secretome of MSCs, as pro-regenerative drugs in in vivo models [https://www.rjpbcs.com/pdf/2017_8(1)/[290].pdf; doi: 10.1186/s13287-019-1479-3, 10.1002/cbin.12222, 10.1038/s12276-023-01017-w]. To strengthen our position, we supplemented the manuscript with recently obtained results of the analysis of the neuroprotective activity of the secretome of immortalized MSCs in an in vitro 3D model of mouse sensory ganglion neurite growth (see sections 2.3.3 and 4.8.3). Moreover, the pro-regenerative (neuroprotective) activity of the secretome of immortalized MSCs was previously confirmed by us in a model of intracerebral hemorrhage in rats [doi: 10.3390/ijms26146697], the study that was cited in this manuscript. We believe that all of this together allows us to assert that the secretome of immortalized MSCs has pro-regenerative activity. In this regard, we would like to leave the title of the manuscript “Safety and Regenerative Properties of Immortalized Human Mesenchymal Stromal Cell Secretome” unchanged.

  1. Immunofluorescence images in Figure 4 should be presented without phalloidin staining, which hinders the observation of secretomes’ effect on alpha-SMA expression.

For separate visualization of alpha-SMA and total actin expression (phalloidin staining), we have added a figure to the Supplementary materials (Figure S2). It shows that extracellular vesicles from primary and immortalized MSCs suppress TGFb-induced alpha-SMA expression.

  1. In figure 5A, a positive Western control of TERT should be presented. Also, tubulin gel is of low quality.

Yes, we agree with you, Figure 5A has been replaced with a more recent one. It still does not show the catalytic subunit of telomerase (TERT) in the 100-fold concentrated iMSC secretome, but it does show alpha-tubulin. The effectiveness of antibodies to TERT is confirmed by the presence of specific bands in the lysates of immortalized MSCs (Figure S4).

Reviewer 3 Report

Comments and Suggestions for Authors

In the current study the authors effectively identify and investigates a few key research gaps. The manuscript addresses an important translational safety question on function or safety of MSC secretome function or safety being affected by whether hTERT-immortalization. The authors have opted for comparisons across multipronged approaches including proteomics, functional readouts, telomerase detection, soft-agar, RNA-seq and have successfully reported extensive proteome overlap (~95%) between primary and immortalized MSCs, suggesting preserved functionality. However, there are few critical points that needs to be addresses before publication.

  1. Throughout the manuscript there is ambiguity between EVs and “secretome,” as methods repeatedly conflate EVs and total secretome. Also, the EVs used for the study are collected using non-standard EV isolation method using 1000-kDa MWCO ultrafiltration only with no EV characterization. Include specific EV characterizations including particle counts/size distribution (NTA), positive/negative protein markers (CD9/CD63/CD81, TSG101and negative controls) purity in compliance with MISEV2023.
  2. The conditioning protocol seems excessively long as secretome collected after 7 days serum-free culture might be associated with risk of cell death and contamination by intracellular proteins. The authors need to include controls for cell viability, LDH release, and secretion normalized to viable cells/time. Otherwise inflated SASP protein level may reflect stress rather than physiology.
  3. The telomerase detection done on the basis of western blot and normalized against tubulin is not an appropriate loading reference for acellular conditioned media and a potential cell-lysis contaminant. Also rhe authors are strongly suggested to provide telomerase activity assay (TRAP) data.
  4. The study reports underpowered transcriptomics. The RNA seq data with n=2 replicates; claiming 99.7% unchanged genes are statistically weak. The study needs to be performed with ≥3 replicates, FDR correction, GEO deposition.
  5. Proteomics data reported elaborates conditioning protocol inconsistent with other assays. Also replicates and raw data not are not provided. The authors are required to include PRIDE deposition, QC, and statistics.
  6. The functional assays also have some discrepancies that need to be addressed. ELISA/qPCR are mostly performed with n=3 which is statistically underpowered [ some acknowledged discrepancies (e.g., HGF) attributed to “small sample power”]. Leydig-cell assay lacks dose–response and defined potency units. Soft-agar assay doesnot consider transformed-cell positive controls. In context to colony counting, justify colony threshold, and quantify colonies per well with blinded counting.

Author Response

Dear Reviewer, Thank you very much for your time and valuable comments! We believe that the manuscript has become much better. Here below, we provide a point-by-point response to your comments. All changes to the original manuscript are highlighted in green.

  1. Throughout the manuscript there is ambiguity between EVs and “secretome,” as methods repeatedly conflate EVs and total secretome. Also, the EVs used for the study are collected using non-standard EV isolation method using 1000-kDa MWCO ultrafiltration only with no EV characterization. Include specific EV characterizations including particle counts/size distribution (NTA), positive/negative protein markers (CD9/CD63/CD81, TSG101and negative controls) purity in compliance with MISEV2023.

Ultrafiltration using the filter with 1000 kDa cut-offs for EV concentration is included into the recommendations made in MISEV2023, section 4 [doi: 10.1002/jev2.12404]. The isolated EVs produced by human adipose-tissue MSCs were previously characterized by several methods according to MISEV2023 recommendations and published in our recent papers [doi: 10.1038/s12276-023-01017-w; 10.3390/cells9051272]. This information was added to the manuscript.

We did not conduct a full characterization of the properties of extracellular vesicles (EVs) in this study, as the focus of our research was different. In this study, we relied on the results previously obtained by our colleagues [doi 10.3390/cells13100861, 10.1155/2020/1289380, 10.1101/2024.12.05.626964], who thoroughly compared the properties of EVs obtained from primary and TERT-immortalized MSC cultures, and showed their high similarity in size, composition, and properties. The results of these studies established that immortalization using TERT does not critically affect the composition and properties of EVs produced by MSCs. In this study, we obtained EVs using a well-established and previously published protocol [10.1038/s12276-023-01017-w; 10.3390/cells9051272]. Using NTA in this study, we confirmed that the secretomes of TERT-immortalized MSCs contain EVs that are similar in size and concentration to those of primary MSCs. These data were added to the materials and methods, results, and supplementary materials of the manuscript.

  1. The conditioning protocol seems excessively long as secretome collected after 7 days serum-free culture might be associated with risk of cell death and contamination by intracellular proteins. The authors need to include controls for cell viability, LDH release, and secretion normalized to viable cells/time. Otherwise inflated SASP protein level may reflect stress rather than physiology.

We have been using this method of culturing and conditioning of adipose tissue-derived MSCs to obtain secretome for a long time and have not observed any significant cell death during conditioning for 7 days. A detailed justification of the MSC conditioning protocol for 7 days, with confirmation of MSC cell viability and analysis of the content of individual factors using the ELISA method, can be found in our previously published article [doi: 10.1038/S12276-023-01017-W]. This information has been added to the manuscript.

We did not measure the concentrations of proteins that mark cell death (e.g., LDH), but we will take your comment into account and conduct this study in the future when using the prolonged conditioning of MSC cultures. We assume that the increase of SASP proteins in the secretome during the late passages is specific and not due to culture aging due to prolonged conditioning, as it is observed only in later passages in pMSCs and iMSCs (and not for all passages of iMSC cultures). For primary MSCs, the increase in SASP at these passages correlates with previously published data [10.1089/ars.2015.6359, 10.18632/oncotarget.7690]. This comment has been added to the manuscript.

  1. The telomerase detection done on the basis of western blot and normalized against tubulin is not an appropriate loading reference for acellular conditioned media and a potential cell-lysis contaminant. Also rhe authors are strongly suggested to provide telomerase activity assay (TRAP) data.

Figure 5A was replaced with a similar one recently obtained. The MSC secretome was concentrated 100-fold, which made it possible to detect even intracellular proteins (alpha-tubulin, the content of which in the secretome is extremely low), but not telomerase. The detection of alpha-tubulin in WB is not so much a normalization as a demonstration of the sensitivity of the method. Additionally, the normalization of the sample load was calculated and controlled by total protein. We added this information to the description of materials and methods in the manuscript. Yes, we understand that the ideal option would be to additionally evaluate the telomerase activity of the concentrated iMSC secretome, but within the time frame allotted for revising the manuscript, it is not possible. Only the synthesis of labeled oligonucleotides in Russia takes at least 3 weeks. Thank you for your recommendation! We will definitely check this in the continuation of our research. We have indicated this limitation of our study in the last two paragraphs of the Discussion section.

  1. The study reports underpowered transcriptomics. The RNA seq data with n=2 replicates; claiming 99.7% unchanged genes are statistically weak. The study needs to be performed with ≥3 replicates, FDR correction, GEO deposition.

Yes, we completely agree with you. Unfortunately, limited funding for this study did not allow us to conduct more repetitions. Repeating this study would take at least two months, which is not possible within the time allotted for responding to Reviewers' comments. We ourselves understand this limitation of our study and honestly point it out in the last paragraph of the Discussion section. The sequencing results were deposited in the GEO repository (https://www.ncbi.nlm.nih.gov/geo/query/acc.cgi?acc=GSE306748) in accordance with the recommendations, and information about this was added to the text of the manuscript.

  1. Proteomics data reported elaborates conditioning protocol inconsistent with other assays. Also replicates and raw data not are not provided. The authors are required to include PRIDE deposition, QC, and statistics.

The use of an alternative conditioning protocol for obtaining samples for proteomic analysis is due to the need to reduce contamination of the samples with MSC degradation products, which inevitably appear in the secretome during prolonged conditioning. Conditioning for 24 hours reduces such contamination, which we have studied in detail in previously published articles [doi: 10.1093/EHJCI/EHAA946.3611; 10.3390/ijms25010290]. The concentrations of therapeutic proteins (growth factors) obtained with this conditioning are extremely low and require a 100-fold or greater concentration of the secretome for their detection or for revealing a noticeable therapeutic effect. For this reason, conditioning for 24 hours was used only for sample preparation for proteomic analysis, but not for in vitro/in vivo tests in this and other studies. This information was added to the manuscript to give readers a better understanding of the logic behind the study.

The raw data of the proteomic analysis were deposited in the PRIDE repository (https://www.ebi.ac.uk/pride/archive/projects/PXD067843) in accordance with the recommendations, and information about this was added to the text of the manuscript. Limited funding for this study did not allow for a large number of replicates (n = 1), which prevents us from performing statistical analysis and quantitatively assessing changes in the expression of a wide range of proteins. However, the data obtained allow us to conclude that the qualitative composition of the MSC secretome is preserved after immortalization. The quantitative analysis of expression of certain growth factors and cytokines was performed using ELISA and qPCR. This information was specified in the text of the manuscript and in the limitations of the study.

  1. The functional assays also have some discrepancies that need to be addressed. ELISA/qPCR are mostly performed with n=3 which is statistically underpowered [ some acknowledged discrepancies (e.g., HGF) attributed to “small sample power”].

Yes, we agree with you that for a number of tests, especially those involving multiple comparisons, it is better to use a larger number of measurements. However, the time allocated for revising the manuscript does not allow us to repeat this experiment. We understand this limitation of our study and honestly point it out in the last paragraph of the Discussion section.

Leydig-cell assay lacks dose–response and defined potency units.

This model for evaluating the biological activity of MSC secretome has been described in detail by us previously [doi: 10.3390/ijms23169414], including an assessment of dose dependence and determination of potency units (a change in the secretory activity of Leydig cells of ≥ 15% indicates the biological activity of the secretome). This test is used to evaluate the pharmacological activity of the secretome as part of a clinical study in the field of regenerative medicine for the treatment of male infertility (No. NCT06869863, https://clinicaltrials.gov/study/NCT06869863). The data obtained served as the basis for registering the patent “Assay for evaluating the secretory activity of model cells” #WO2024177532A1 (https://patents.google.com/patent/WO2024177532A1/en). This information was added to the manuscript. Therefore, this study uses a previously validated model to evaluate the biological activity of the secretome of immortalized MSCs.

Soft-agar assay doesnot consider transformed-cell positive controls.

In this study, we do not use initially transformed cell cultures, but still we have two positive controls: primary fibroblasts cultures treated with dimethyl sulfate or sodium nitrite, substances with prominent transforming (carcinogenic) activity. When these substances are added, we observe the formation of a certain number of multicellular colonies, which indicates that the conditions allow the cells to grow in soft agar (which is the function of the positive control in this case) if they are able to grow in the absence of contact-mediated signals with the matrix and other cells.

In context to colony counting, justify colony threshold, and quantify colonies per well with blinded counting.

Yes, the researcher who counted the colonies and processed the raw data was blinded. The threshold for exerting the potential transformative activity was considered to be the appearance of at least one multicellular colony. This information has been added to the manuscript. Such colonies were absent in the experimental groups (treated with pMSC or iMSC secretomes) but were present in the positive control groups (treated with dimethyl sulfate or sodium nitrite), demonstrating the validity of this test and the absence of transforming activity of the secretomes of primary and immortalized MSCs.

Round 2

Reviewer 1 Report

Comments and Suggestions for Authors

The Authors have implemented important changes to improve the manuscript structure and clarity, especially regarding introduction, rationale for experiments, and limitations.

However, there are still some points that need to be addressed:

  1. In the figure legend of the Figure 6a and b, the authors state: “No statistical analysis was performed, it is a qualitative test.” While I agree that statistics are not applicable to qualitative yes/no outcomes, it is essential to demonstrate reproducibility. Reporting a result from a single experiment is not sufficient, qualitative assay should be repeated at least three independent times. In the figure legend it should say: Representative result of n independent experiments is presented.
  2. The number of replicates should also be at least 3 in Figure 4.
  3. The quality and consistency of the figures need improvement. The graphs vary in size and formatting, which detracts from clarity and professionalism. All graphs within a manuscript should have the same size, font style, and general layout.

    4. Beyond aesthetics, there are also issues with axis scaling. When presenting comparable results in the same figure, the y-axis numbering should be consistent. All graphs should use the same scale with identical tick marks (for example, 0-5, steps of 1). This ensures that comparisons are meaningful and prevent visual misinterpretation. This refers to Figures 1-3.

Author Response

Dear Reviewer, Thank you very much for your time and valuable comments! We believe that the manuscript has become much better. Here below, we provide a point-by-point response to your comments. All changes to the previously revised manuscript are highlighted in yellow.

The Authors have implemented important changes to improve the manuscript structure and clarity, especially regarding introduction, rationale for experiments, and limitations.

However, there are still some points that need to be addressed:

1. In the figure legend of the Figure 6a and b, the authors state: “No statistical analysis was performed, it is a qualitative test.” While I agree that statistics are not applicable to qualitative yes/no outcomes, it is essential to demonstrate reproducibility. Reporting a result from a single experiment is not sufficient, qualitative assay should be repeated at least three independent times. In the figure legend it should say: Representative result of n independent experiments is presented.

There is a misunderstanding. Of course, this experiment (Fig. 6A) was repeated several times (n = 4), and I have included the results of all of these independent experiments in the Fig. S4 (Supplementary materials). Each of them revealed that the amount of telomerase in the 100-fold concentrated iMSC secretome is below the limit of quantification. The information about the number of independent repeats has been added to the manuscript (Materials and Methods) and the caption for Figure 6.

The data presented in Figure 6B was obtained during two independent experiments (two cultures of primary human dermal fibroblasts from different donors), each of which was performed in triplicates. This information was added to the manuscript (Materials and Methods) and Figure 6 caption. Each experiment yielded similar results: multicellular colonies were observed only in the positive control groups treated with NaNO2 and DMS, but not in the pMSC and iMSC secretome groups. This test is qualitative - it gives the answer “yes” or “no”. The appearance of even a single multicellular colony in the experimental group indicates the presence of potential transforming activity of the drug candidate, that must be thoroughly further studied. In this experiment, no transforming activity was established for pMSC and iMSC secretomes.

2. The number of replicates should also be at least 3 in Figure 4.

We have just received new data, which we have added to the sample mentioned above. The data obtained confirm the results obtained earlier. Figure 4, its caption, and the information in the text have been updated.

3. The quality and consistency of the figures need improvement. The graphs vary in size and formatting, which detracts from clarity and professionalism. All graphs within a manuscript should have the same size, font style, and general layout.

Thank You for your comment! We tried to consider your recommendations as much as possible. Yes, sometimes graphs may vary slightly in size, which is due to the fact that they are part of a complex figure that includes microscopic, histological, biochemical, and other data.

4. Beyond aesthetics, there are also issues with axis scaling. When presenting comparable results in the same figure, the y-axis numbering should be consistent. All graphs should use the same scale with identical tick marks (for example, 0-5, steps of 1). This ensures that comparisons are meaningful and prevent visual misinterpretation. This refers to Figures 1-3.

Thank You for your comment! We tried to consider your recommendations as much as possible. In Figure 2, the original Y-axis scale was retained to make the differences between the groups (pMSC and iMSC) more noticable. This is justified because each of the three graphs in Figure 2 shows the content of different molecules: MCP-1, PAI-1 or IL-6, which are produced at different levels.

Reviewer 2 Report

Comments and Suggestions for Authors

The authors properly answered to the critiques

Author Response

Dear Reviewer, Thank you very much for your time and valuable comments! 

Reviewer 3 Report

Comments and Suggestions for Authors

The authors have concisely answered all the queries and improved the manuscript accordingly. Should be accepted after thorough text editing

Author Response

(The authors gave the same response as above.)

Round 3

Reviewer 2 Report

Comments and Suggestions for Authors

The authors properly answered the critiques

Author Response

(The authors gave the same response as above.)
